# ⟳ InfCycle: Learning to Use Tools via Inference Compute and Cycle Consistency

## Abstract

The scaling of inference-time computation in large language models (LLMs) has emerged as a promising approach for enhancing reasoning capabilities by trading off inference-time and pre-training compute. The practice of how to enable LLMs to utilize additional computation at test time to improve response accuracy is crucial for both academia and industry. *Proposer-Verifier*, as a typical paradigm of inference scaling, often fails to generalize to various scenarios. Specifically, in tool use tasks, LLM face the risk of lacking effective verifiers, leading to error accumulation across multiple reasoning steps. In this work, we address these challenges by introducing **InfCycle**, a multi-stage data synthesis strategy that employs LLMs as a data synthesizer and cycle consistency verification to ensure high-quality trajectory generation. This approach utilizes step-wise cycle consistency among synthesized trajectories for a given tool, providing effective process supervision that has advantages over outcome supervision. Extensive experiments on multiple tool use and reasoning tasks demonstrate that InfCycle efficiently enables self-improvement. It outperforms state-of-the-art baselines on StableTool-Bench, achieving a 75.4% pass rate and a 79.6% win rate using small models (7B), without relying on external supervision or expert trajectories for warm-up.

## 1 Introduction

Tool use is a critical capability for LLMs, enabling them to perform complex reasoning tasks through multi-step inference, and interact with real-world environments (Mallen et al., 2022; Wang et al., 2023b; Zeng et al., 2023; Xu et al., 2023b; Huang et al., 2024). While many studies have focused on the challenges of tool use, they often rely heavily on *imitation learning* (Hussein et al., 2017), such as learning from teacher models (Yang et al., 2024; Qin et al., 2023a) (the OpenAI GPT series (Achiam et al., 2023)) or human-annotated execution trajectories. Moreover, tool invocation scenarios require LLMs to perform very complex multi-step reasoning, posing a great challenge to ensure the accuracy of intermediate steps, especially when lacking of reliable verification. Consequently, the high cost of data synthesis in imitation learning and the requirements for reliable verifiers hinder the progress of tool use by LLMs. This research focuses on exploring **how LLMs can autonomously learn tool use capabilities from scratch, without relying on any external supervision.**

Recently, leveraging inference-time compute (Brown et al., 2024; Wu et al., 2024; Chen et al., 2024a) to enhance model capabilities has become an effective method for enhancing model capabilities and facilitating self-improvement. The *Proposer-Verifier* (Snell et al., 2024) strategy is a common method that increases the sampling cost of the LLM and selects the highest score according to a verifier, thereby trading off inference cost for performance improvement. However, in most tool use scenarios, the **lack of a reliable verifier** undermines the effectiveness of inference-time scaling (Liu et al., 2024c; Mekala et al., 2024; Liu et al., 2024a; Ye et al., 2024). Additionally, in multi-step reasoning scenarios, **error accumulation** may hinder small size models from identifying the correct execution path, significantly reducing the effectiveness of data sampling.

To address these issues, we first decompose the tool use process into multi-step reasoning tasks, enabling models with limited capabilities to be integrated into our data synthesis pipeline. Crucially, we develop two modules for data synthesis: the *Generator* and *Simulator*, which generate intermediate steps for tool invocation as training data for the LLM. As illustrated in Figure 1, the principle of cycle consistency (Zhu et al., 2017) ensures that inconsistent intermediate steps are less likely to

Figure 1: The *Generator* can leverage the LLM to create potential user queries based on the available API information. The *Simulator* can interact with the environment using these queries to achieve API response results. The cycle consistency constraint between them can extract high-quality synthetic execution trajectories in the absence of explicit human supervision.

form reasonable execution trajectories, significantly reducing the effects of LLM hallucinations (Ji et al., 2023). Since this approach does not rely on additional supervision, it **serves as an efficient verifier**, enabling scalable data generation.

Furthermore, we need to tackle the challenges of error accumulation and long-distance tool reasoning, which are caused by model hallucinations and inherent limitations. Frequent trial-and-error processes are not suitable for data synthesis, leading to inefficiencies and hindering scalability. Thus, we propose InfCycle, **a multi-stage synthesis strategy** that alternates between model inference and training to achieve self-improvement. We integrate the $A^\star$ search algorithm (Zhuang et al., 2023) to enhance sampling efficiency and employ preference learning algorithms to improve the model's generalization ability (Chen et al., 2024b). This strategy progressively addresses the following problems: 1) the model's inability to handle JSON-formatted context and outputs (Yuan et al., 2024); 2) inefficiencies in the *Simulator* when searching for execution trajectories from the *Generator*; and 3) the LLM's tendency to accumulate errors during multi-step execution. In summary, our approach ensures two critical factors in data scaling: the **Precision** and **Coverage** (Brown et al., 2024) of correct execution trajectories, effectively unlocking the potential of inference-time computation.

Our main contributions are as follows:

- In tool use scenarios, we investigate how to effectively leverage additional computation at test time to enhance accuracy, enabling LLMs to improve their capability through self-improvement from scratch.
- We propose InfCycle, which employs a multi-stage trained model to act as a *Proposer*, and utilizes a cycle consistency mechanism to serve as a *Verifier*. Overall, this approach significantly boosts data sampling efficiency and model performance.
- We validate our method using multiple models with varying capabilities on the StableToolbench, such as Qwen2.5-7B, which surpasses GPT-4, achieving a 75.4% Pass Rate and a 79.6% Win Rate, as well as the Berkeley Function Calling Benchmark, where Meta-LLaMA3-8B achieve an improvement of over 16.31 points. Additionally, extensive experimental results and analyses confirm the effectiveness and scalability of our approach.

## 2 RELATED WORKS

**Tool Use**    As pioneers, Toolformer (Schick et al., 2024), Gorilla (Patil et al., 2023), and ToolAlpaca (Tang et al., 2023) have explored the potential of LLMs in tool use. ToolLlama Qin et al. (2023b) notably expanded the number of available tools, exceeding 10,000 APIs, and investigated the possibilities of data scaling. Many related works primarily seek improvements through two approaches: **Inherent Abilities**: This involves manipulating prompts or enhancing the execution framework. Xu et al. (2023b) utilize examples, in-context demonstrations, and generation styles to explore the potential of LLMs. AutoAct (Qiao et al., 2024) employed a multi-agent collaboration framework to complete reasoning tasks. RestGPT (Song et al., 2023) introduced a coarse-to-fine online planning mechanism by using three main modules (Planner, API Selector, and Executor). **Synthetic Data**: This strategy empowers model capabilities through synthetic data. ToolVerifier (Mekala et al., 2024) leveraged the LLaMA-2 70B model to verify the accuracy of synthetic data. APIGen (Liu et al., 2024c) used a strong model to filter API calls based on rules and semantics, ensuring data accuracy. In contrast, our approach combines the strengths of both methods, and our setups do not rely on any teacher LLM or external supervision.

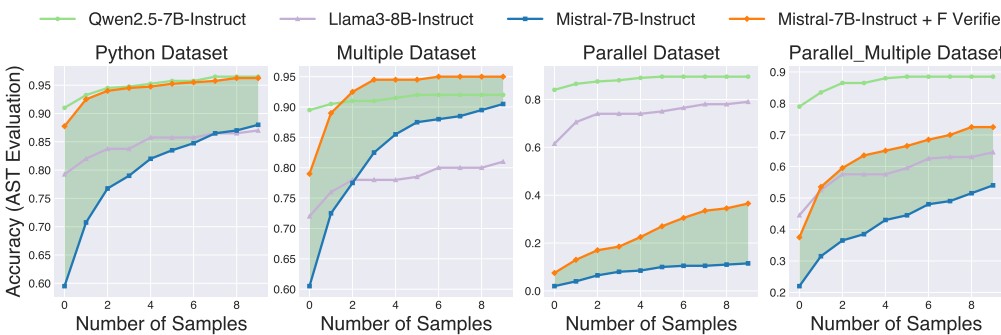

Figure 2: The figure shows experiments on the BFCL, comparing accuracy curves with limited sampling ($< 10$). The green area indicates performance improvement via Format Verifier $\mathcal{V}_f$.

**Inference Scaling**   LLMs can utilize techniques such as CoT (Wei et al., 2022) or Reflection Shinn et al. (2024) to enhance their reasoning capabilities during testing. However, many studies show that these methods often have limited effectiveness for complex tasks (Huang et al., 2023; Stechly et al., 2023; Valmeekam et al., 2023). Nevertheless, this research direction remains crucial for the future, particularly in exploring the trade-offs between inference time and pre-training computing. Brown et al. (2024) demonstrate that scaling inference computing through repeated sampling leads to significant improvements in coverage across various tasks and models. Snell et al. (2024) introduce a compute-optimal strategy that enhances the efficiency of test-time compute scaling compared to a best-of-N methods. In contrast, we are the first to employ trajectory cycle consistency in tool use to construct a verifier, thereby mitigating the requirements for inference precision.

## 3   MOTIVATIONS

**Problem Formulation**   In tool use scenarios, the LLM receives a user query $\mathcal{Q}$ along with a set of candidate API functions, represented as $\mathcal{A} = \{\texttt{API}_0, \texttt{API}_1, \ldots, \texttt{API}_{|\mathcal{A}|}\}$. The goal of the LLM is to fulfill the user's intent by executing a specific sequence of API function calls. The decision process can be described as $y \sim \pi(y|s_0, a_1, a_2, \cdots)$, where $\pi(\cdot)$ represents the policy, $s_0$ denotes the initial task state, and $a$ represents the actions taken by the model, such as selecting or executing a specific API function from $\mathcal{A}$. Each action may update the state $s_i$, guiding subsequent decisions.

**Preliminary Experiments**   We conduct investigations to evaluate the impact of increasing reasoning steps on the Berkeley Function-Calling benchmark. Specifically, we use the LLM as a *Proposer* and combine it with two *Verifiers* [1] to explore inference-time scaling: the *Correct Verifier* $\mathcal{V}_c$, which checks if the model output matches the expected correct answer, and the *Format Verifier* $\mathcal{V}_f$, which ensures that the output conforms to JSON format or the specified API parameter types, for example, distinguishing between numeric types such as *floats* and *integers*.

Specifically, we sample multiple outputs using the LLM as $\pi(\cdot)$ and select the results that pass the verification process: $y^* = \arg\max_{y_i \in \{y_1, y_2, \ldots, y_m\}} \mathbb{I}[\mathcal{V}_c(y_i) = 1]$. Here, $y_i$ represents the generated results obtained through sampling. For both LLaMA3-8B-Instruct and Qwen2.5-7B-Instruct, we conduct experiments using their official inference frameworks [2]. As illustrated in Figure 2, the LLMs demonstrate significant performance improvements across different datasets, suggesting that better data trajectory samples exist within a limited search space for open-source LLMs. In the case of Mistral-7B-Instruct-v0.2, the model struggles to generate JSON formatted outputs, making it nearly impossible to complete user queries on the benchmark. Therefore, we utilize the framework proposed in Section 4.1 for inference. We observe conclusions similar to those previously noted. When we introduce the *Format Verifier* using $\mathcal{V}_c(\mathcal{V}_f(y_i))$, the model's performance significantly exceeds that of using only the *Correct Verifier* $\mathcal{V}_c(y_i)$ alone.

---

[1]**Note: The verifier uses test set ground truth to validate outputs and is not applied in other sections**

[2]`https://github.com/ShishirPatil/gorilla/tree/main/berkeley-function-call-leaderboard`

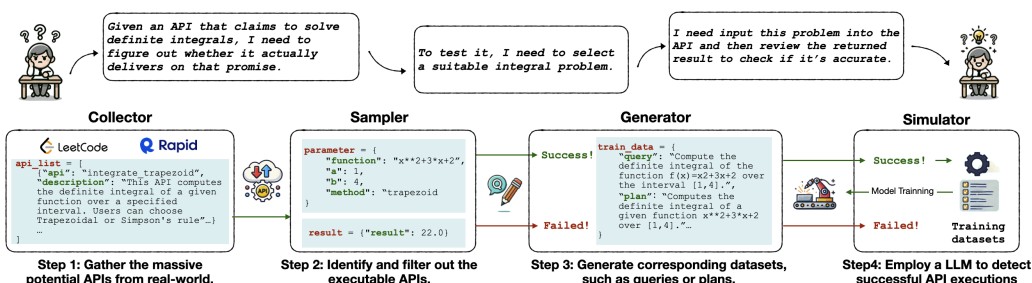

Figure 3: The data synthesizer can sequentially run four modules to transform JSON-formatted API information into multi-step execution trajectory data for model training. This process is designed to mimic how humans learn to use APIs: individuals first hypothesize that an API can solve a specific problem and then validate this hypothesis by executing the API.

**Critical Insight**    From the experiments, we can draw the following key conclusions: (1) Increasing the inference cost leads to significant improvements in model performance for both single-step and multi-step reasoning tasks, regardless of the model's capabilities. (2) Task decomposition simplifies complex problems, allowing models to tackle previously challenging issues, as shown by Mistral's poor performance without decomposition. (3) Our proposed reasoning framework, combined with the *Format Verifier* $\mathcal{V}_f$ for process validation, enables small-size models to outperform stronger models in certain tasks. These findings indicate that **process verifiers are more effective than solely outcome verifiers in enhancing long-range reasoning capabilities and assisting models in identifying high-quality execution trajectories.**

# 4    METHODS

In this section, we first build a holistic data synthesis pipeline to generate training data of tool invocation trajectories, as shown in Figure 3. Then, we discuss how to utilize cycle consistency to filter the data synthesized by the *Generator* and *Simulator*. Finally, we introduce InfCycle, a multi-stage synthesis strategy designed to help LLMs learn the ability of tool use from scratch.

## 4.1    THE PIPELINE OF DATA SYNTHESIZER

To validate data scalability in large-scale tool use, we select real-world APIs as our tool candidates. To support this, we create and deploy an execution environment for these APIs, enabling type checking and providing execution feedback. For more details, please refer to Section 5.

**Collector**    The execution environment naturally acts as a verifier, ensuring that the API request can be executed successfully while filtering out invalid requests. We collect the available APIs as tool candidates from the environment and maintain input consistency through a unified format.

**Sampler**    Considering the complexity of user intent, multiple tools are often needed to achieve a solution. Thus, we categorize the tool collection into three types: *Simple*: Users utilize a single tool. *Parallel*: Users make multiple calls to the same tool. *Multiple*: Users employ multiple tools. The sampled tool cases include 'API definitions', 'request parameters', and 'execution results', which we use to build tool-invocation trajectory data.

**Generator**    Based on the sampled API execution parameters and results, we can use LLMs to generate potential user queries $\mathcal{Q} = \{q_1, q_2, \cdots, q_{|\mathcal{Q}|}\}$. To ensure accuracy, we use LLMs as semantic checkers (Liu et al., 2024c). They filter out unsuitable examples by evaluating whether the user query originates from the real world and whether the response adequately meets the user's intent. For each example, we construct execution paths for task planning based on the order of calls. Specifically, we obtain synthesized action trajectories $\{q, \hat{a}_p, \hat{a}_s, \hat{a}_e\}$, where $\hat{a}_p$ represents task planning, $\hat{a}_s$ denotes the selected API and parameters, and $\hat{a}_e$ indicates the execution results.

Table 1: The table shows two types of errors that are present in the filtered samples.

| Mistake | Content |
|---|---|
| *Internal Logical Error* | *Query*: I have an equation that describes a signal over time, given by $x(t) = Re\left(Ae^{j\pi Bt}\right) + Re\left(De^{j\pi Et}\right)$, where A and D are constants. If A=3, D=0.5, *E=5, and B=4*, what are the *values of B and E* in this equation? |
| *Planning Error* | *Sub-Plan*: Since find_triplet_equal_sum is supposed to solve the user's query directly, *skip calling* the twosum function for now. |

**Simulator**  To adapt models with limited capabilities, we decompose the reasoning process of tool invocation into five steps: 1) *Task Planning*: The model starts by analyzing the user query $q$ and decomposing it into sub-tasks $a_p$, similar to ReWoo (Xu et al., 2023a). 2) *Tool Selection*: Based on the planned sub-tasks, the model takes action $a_s$ to choose the appropriate tools and parameters. 3) *Tool Execution*: The model then interacts with the environment to gather results $a_e$; 4) *Tool Reflection*, Through action $a_r$, the model evaluates execution feedback to determine if the sub-plan is complete or if the API needs re-execution. 5) *Task Reflection*, Finally, the model uses $a_{tr}$ to assess if the task is completed and, if necessary, re-planed based on its execution history. Through the aforementioned steps, we derive the sequential action trajectory $\{a_p \to a_s \to a_e \to a_r \to a_{tr}\}$, respectively. For simplicity, we omit the explicit states on which the model's predictions are based. However, these states are implicitly derived from the outcomes of previous actions, such as $a_s \sim \pi(s_0, a_p)$ or $a_r \sim \pi(s_0, a_p, a_s, a_e)$. More inference details refer to Appendix A.5.

This multi-step execution process effectively addresses two key challenges in model reasoning: First, it bridges the gap between natural language and JSON format. Many general models struggle to handle JSON context (Tam et al., 2024), which can impair their performance on tasks that require this format. Second, it simplifies decision-making processes. In methods like ReACT (Yao et al., 2022), the model must choose the next action from multiple options, which can be challenging for models with limited reasoning abilities.

### 4.2 THE STEP-WISE CYCLE CONSISTENCY

Given candidate APIs, the *Generator* filters out semantically incorrect trajectories, while the *Simulator* excludes user queries that cannot be executed successfully. Furthermore, we verify the cycle consistency of trajectories between $\{\hat{a}_p, \hat{a}_s, \hat{a}_e\}_i^S)$ and $\{a_p, a_s, a_e\}_i^S$, where $S$ represents the total number of reasoning steps. We ensure that each action in the multi-step reasoning process is consistent and executed sequentially.

**Why is the cycle consistency verifier effective?**  The *Generator* may overlook numerous errors, even when selecting the prompt carefully. These issues are unpredictable, as illustrated in Table 1: 1) **Internal logical issues within the user query**: These issues can lead to synthesized queries that lack executable trajectories. 2) **Semantic problems in planned sub-tasks**: Such problems can result in synthesized trajectories that are fundamentally unreasonable. Unlike recent approaches (Qin et al., 2023b; Liu et al., 2024c) that use an **outcome verifier** to filter samples, step-wise cycle consistency as a **process verifier** ensures the accuracy of the reasoning process, thereby guaranteeing the high quality of the synthesized data.

### 4.3 MULTI-STAGE SYNTHESIS STRATEGY

$A^\star$ **search**  We define the entire reasoning process starting from $s_0$ as an expansion into a decision tree $\mathcal{T}$, with all actions being represented as nodes $\mathcal{V}(\mathcal{T})$. The successors of each node are generated by using temperature sampling. Specifically, we first select a node $n$ from the frontiers of the tree (denoted as $\mathcal{F}(\mathcal{T}) \subseteq \mathcal{V}(\mathcal{T})$) according to the cost function. Then, we expand node $n$ using the LLM to generate $k$ subsequent actions, which are used to update $\mathcal{F}(\mathcal{T})$.

The $A^\star$ algorithm aims to search for an appropriate path that minimizes the cost function $c(n) = g(n) + h(n)$, where $n$ is the current node, $g(n)$ represents the cost of the path from the start node to $n$, and $h(n)$ is a heuristic function estimating the cost of the cheapest path from $n$ to the goal. In practice, we define $c(n)$ as the total number of reasoning steps required to complete the user query,

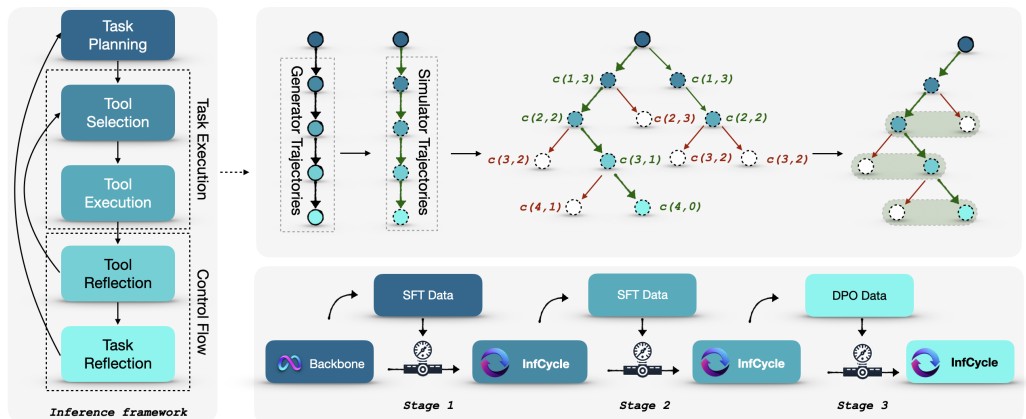

Figure 4: **Left**: The inference framework decomposes the tool use task into a multi-step reasoning process, where each step represents a distinct action. **Right**: Each circular node corresponds to a specific reasoning step, visualizing actions as part of the overall decision-making process. We leverage both Chain and Tree structures for reasoning (the $c(h(n), g(n))$ represents cost function). This tree-based approach also facilitates the creation of pairwise preference data for learning tasks.

$g(n)$ as the number of steps executed so far, and $h(n)$ as the remaining steps needed to reach the goal. To simplify the design of the heuristic function, we use the number of remaining elements obtained by subtracting the intersection of the trajectory sets generated by the *Generator* from the total set. Considering the issue of diversity in temperature sampling, we enhance the diversity of candidates in the $A^\star$ algorithm by adding a beam. Specifically, we select $k$ nodes (default $k = 2$) from $\mathcal{F}(\mathcal{T})$ as candidates for expansion in each iteration.

Using $A^\star$ search serves two purposes: 1) It enables quicker identification of a reasonable path by utilizing reference paths provided by the *Generator*, which guide the search process towards viable solutions more efficiently, As shown in Figure 5, 2) It facilitates the synthesis of DPO data, as pairwise data relies on a consistent context to ensure suitable comparisons.

**Synthesis and learning** We apply the simulator for data synthesis in three stages. **Stage 1**: We use an open-source instruct model to interact with the environment and generate chain of thought trajectories. **Stage 2**: Leveraging the Stage 1 synthesized data, we train an initial model and use a tree-search method to produce new trajectories, filtering out unreasonable samples through cycle consistency. This iterative process facilitates effective data scaling. **Stage 3**: From the tree structure's synthesized trajectories, we compare sibling nodes to identify correct and incorrect pairs for preference learning training data. We then apply Direct Preference Optimization (DPO) to enhance the model's ability to more effectively distinguish between competing trajectories.

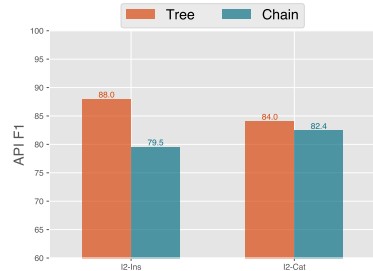

Figure 5: The figure compares API F1 Scores on the testset using Tree-search and CoT for different tool invocation.

## 5 EXPERIMENTS

We chose StableToolBench (Guo et al., 2024) and the Berkeley Function-Calling (BFCL) (Yan et al., 2024) benchmark to evaluate the effectiveness of our proposed method. StableToolBench requires real-time interaction with the RapidAPI [3] to gather feedback, primarily evaluating the model's performance in a dynamic environment. In contrast, BFCL employs a static evaluation set that emphasizes the model's ability to extract complex APIs and parameters. To achieve this, we collected 6k APIs from RapidAPI and converted 2k code problems from LeetCode into usable APIs. For additional statistics and experimental details, please refer to the Appendix A.

---

[3] https://rapidapi.com/hub

Table 2: We calculate the pass rates (%) by averaging the results of each model over three trials. All evaluations are conducted using GPT-4 Turbo, following official guidelines, to ensure comparability.

| Method | Model | Inf | I1-Ins | I1-Cat | I1-Tool | I2-Cat | I2-Ins | I3-Ins | Avg. |
|---|---|---|---|---|---|---|---|---|---|
| ToolLLaMA | ∞ | CoT | $51.8_{\pm0.4}$ | $53.1_{\pm0.6}$ | $46.4_{\pm1.2}$ | $51.6_{\pm1.1}$ | $48.9_{\pm0.4}$ | $37.2_{\pm0.8}$ | 48.2 |
| ToolLLaMA | ∞ | DFS | $61.0_{\pm1.8}$ | $58.8_{\pm0.6}$ | $45.6_{\pm1.2}$ | $60.3_{\pm1.1}$ | $53.5_{\pm0.4}$ | $48.1_{\pm0.8}$ | 54.6 |
| GPT4-Turbo | ⑤ | CoT | $52.8_{\pm1.3}$ | $56.6_{\pm0.9}$ | $51.9_{\pm0.5}$ | $51.9_{\pm1.1}$ | $52.8_{\pm0.4}$ | $52.5_{\pm0.8}$ | 53.1 |
| GPT4-Turbo | ⑤ | DFS | $59.2_{\pm0.5}$ | $61.7_{\pm0.7}$ | $65.7_{\pm1.0}$ | $55.6_{\pm0.6}$ | $55.2_{\pm0.4}$ | $52.5_{\pm4.3}$ | 60.6 |
| TP-LLaMA | ∞ | DFS | $55.0_{\pm0.0}$ | $65.0_{\pm0.0}$ | $\mathbf{80.0}_{\pm0.0}$ | $75.0_{\pm0.0}$ | $67.0_{\pm0.0}$ | $61.0_{\pm0.0}$ | 65.0 |
| Tool-Planner | ⑤ | P&S | $66.0_{\pm0.0}$ | $78.5_{\pm0.0}$ | $75.0_{\pm0.0}$ | $\mathbf{83.5}_{\pm0.0}$ | $77.5_{\pm0.0}$ | $\mathbf{83.0}_{\pm0.0}$ | **77.3** |
| Tool-Planner | A\ | P&S | $64.0_{\pm0.0}$ | $77.0_{\pm0.0}$ | $59.5_{\pm0.0}$ | $79.5_{\pm0.0}$ | $76.5_{\pm0.0}$ | $78.0_{\pm0.0}$ | 72.4 |
| **InfCycle** | ⓜ | P&S | $68.6_{\pm0.4}$ | $57.7_{\pm0.4}$ | $44.5_{\pm1.6}$ | $50.3_{\pm0.8}$ | $69.1_{\pm1.9}$ | $63.7_{\pm4.3}$ | 59.0 |
| **InfCycle** | ∞ | P&S | $70.6_{\pm1.8}$ | $69.3_{\pm0.0}$ | $70.7_{\pm0.5}$ | $55.8_{\pm0.6}$ | $71.8_{\pm0.7}$ | $70.8_{\pm1.0}$ | 68.2 |
| **InfCycle** | 🐦 | P&S | $\mathbf{71.2}_{\pm1.4}$ | $\mathbf{80.8}_{\pm0.9}$ | $78.3_{\pm0.4}$ | $63.2_{\pm0.7}$ | $\mathbf{80.8}_{\pm1.2}$ | $77.9_{\pm0.7}$ | 75.4 |

Table 3: The results of win rates (%) for different models are calculated by comparing with the GPT-3.5-Turbo.

| Methods | Model | Inf | I1-I | I1-C | I1-T | I2-C | I2-I | I3-I | Avg. |
|---|---|---|---|---|---|---|---|---|---|
| ToolLLaMA | ∞ | CoT | 41.7 | 45.1 | 32.3 | 52.8 | 46.8 | 26.2 | 40.8 |
| ToolLLaMA | ∞ | DFS | 42.3 | 51.0 | 31.0 | 67.0 | 54.0 | 31.1 | 54.0 |
| GPT4-Turbo | ⑤ | CoT | 71.2 | 77.1 | 61.4 | 79.2 | 71.8 | 67.2 | 71.3 |
| GPT4-Turbo | ⑤ | DFS | 73.0 | 75.2 | 68.4 | 77.4 | 66.9 | 60.7 | 70.2 |
| TP-LLaMA | ∞ | DFS | 56.0 | 59.0 | 54.0 | 70.0 | 64.0 | 86.0 | 65.0 |
| Tool-Planner | ⑤ | P&S | 75.5 | 75.8 | 71.8 | 79.8 | 70.3 | **92.0** | 77.5 |
| Tool-Planner | A\ | P&S | 73.8 | 76.3 | 73.8 | 79.3 | 68.3 | 87.5 | 76.5 |
| **InfCycle** | ⓜ | P&S | 62.0 | 62.1 | 54.4 | 70.8 | 65.3 | 62.3 | 62.8 |
| **InfCycle** | ∞ | P&S | **78.5** | 75.8 | **77.8** | 74.5 | **78.2** | 65.6 | 75.1 |
| **InfCycle** | 🐦 | P&S | 76.1 | **86.9** | 74.1 | **81.1** | 75.8 | 83.6 | **79.6** |

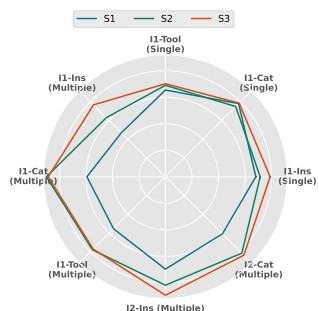

Figure 6: API F1 Score Results.

## 5.1 STABLETOOLBENCH

**Evaluation Setup** In this experiment, **I1** represents intra-category multi-tool instructions, **I2** denotes intra-collection multi-tool instructions, and **I3** includes unseen instructions for the same tools as those in the training data. We categorized unseen tools into three groups: (1) **Ins** for unseen instructions related to the same tools, (2) **Tool** for unseen tools within the same (seen) category, and (3) **Cat** for unseen tools in a different category. We compared the performance of different models based on the official evaluation metrics: **Pass Rate**: This metric measures the proportion of successfully completed instructions within limited budgets, indicating the executability of instructions for LLM. **Win Rate**: This metric involves providing an instruction along with two solution paths to a GPT evaluator, which determines the preferred solution.

**Baselines** We compared several strong baselines. ToolLLaMA Qin et al. (2023b) is trained using distilled data from ChatGPT and uses depth-first tree search (DFS) for reasoning. TP-LLaMA Chen et al. (2024b) employs reinforcement learning through synthesized preference pairs. ToolPlanner Liu et al. (2024b) utilizes a unique scheduling approach, focusing on Plan-and-Solve Wang et al. (2024) (P&S) by organizing tasks before invoking functions. This planning-centric research necessitates models with strong reasoning capabilities, often relying on closed-source models. In contrast, our study synthesizes data using less powerful models to improve reasoning abilities.

### 5.1.1 RESULTS

As shown in Table 2 and Table 3, when interacting with RapidAPI, InfCycle significantly outperforms previous models across six different test sets, achieving higher Pass Rates and Win Rates. With different backbones (Mistral-7B-Instruct-v0.2, LLaMA3-8B-Instruct, and Qwen2.5-7B-Instruct), our approach demonstrates outstanding tool invocation capabilities. Our data synthesis method relies solely on open-source models, with our 7B model outperforming GPT-4-based strategies such as ToolPlanner, illustrating the effectiveness and compatibility of our approach. Although TP-LLaMA also uses an open-source model for its Tree Search algorithm and generates DPO data, merely enhancing the model through preference learning does not fully exploit its tool invocation

Table 4: The model performance on BFCL. InfP represents the Inference Pattern, where FC directly returns JSON, while Prompt requires post-processing for results. Mistral-7B$^*$ uses the inference framework with InfCycle because the official scripts fail to produce effective calls.

| Method | InfP | Model | Abstract Syntax Tree (AST) Evaluation | | | | Avg |
| | | | Simple | Multiple | Parallel | Parallel Multiple | |
|---|---|---|---|---|---|---|---|
| GPT-4o | Prompt | | 73.58 | 92.50 | 91.50 | 84.50 | 85.52 |
| GPT-4o-mini | Prompt | | 79.67 | 89.50 | 89.00 | 88.00 | 86.54 |
| o1-mini | Prompt | | 68.92 | 89.00 | 73.50 | 70.50 | 75.48 |
| Command-R-Plus | FC | | 71.10 | 85.00 | 80.00 | 66.00 | 75.54 |
| Open-Mixtral-8x22 | Prompt | | 50.50 | 95.00 | 8.50 | 70.50 | 56.12 |
| Mistral-7B | Prompt | | 0.70 | 0.00 | 0.00 | 0.00 | 0.18 |
| Mistral-7B$^*$ | Prompt | | 19.83 | 60.50 | 2.00 | 22.00 | 26.08 |
| **InfCycle** | Prompt | | $65.50_{\uparrow 45.67}$ | $79.50_{\uparrow 19}$ | $72.00_{\uparrow 70}$ | $61.50_{\uparrow 39.5}$ | 69.63 |
| Meta-LLaMA-3-8B | Prompt | | 58.53 | 78.00 | 59.50 | 53.25 | 62.32 |
| **InfCycle** | Prompt | | $67.00_{\uparrow 8.47}$ | $90.00_{\uparrow 12}$ | $81.00_{\uparrow 21.5}$ | $76.50_{\uparrow 23.25}$ | 78.63 |

capability. This discrepancy highlights the gap between synthesized data and real data, emphasizing the importance of reliable verifiers to filter data and ensure accuracy.

### 5.1.2 HUMAN EVALUATION

Given that the evaluations in the main results rely on model judgments, which can often be unreliable (Wang et al., 2023a), we enhance accuracy by conducting human annotation of the Stable-ToolBench test set. First, we remove inaccessible and invalid samples by interacting with RapidAPI Website and supplement them with new user queries from the same tool candidates of StableTool-Bench. Next, we categorize the collected samples into two groups: those that can fulfill the user's intent with a **single** API and those that require **multiple** APIs. Finally, we manually annotate the actual execution trajectories for these samples to ensure precision.

We examine InfCycle performance trends across various data synthesis stages (denoted as S1, S2, and S3) using the API F1 score as our evaluation metric. This score evaluates the alignment between predicted and ground truth APIs, showcasing the model's ability to select appropriate tools. We deliberately use a small-size model (Mistral-7B-Instruct) as the backbone to determine if it can gradually synthesize higher-quality data. As Figure 6 illustrates, there is a clear improvement in performance throughout different stages. Notably, the most significant gains appear in testsets requiring multiple APIs because of the initial limitations of the small-size model in handling only simple tasks. Our multi-stage synthesis strategy enables the model to progressively acquire the capability to tackle complex scenarios and tasks.

### 5.2 BERKELEY FUNCTION-CALLING BENCHMARK

**Evaluation Setup** BFCL includes various types of test sets: **Simple**: Involves one API candidate and calls a single function. **Multiple**: Calls one function using 2 to 4 API candidates based on the user query. **Parallel**: Invokes multiple functions simultaneously from a user query. **Parallel Multiple**: Combines the previous two, allowing multiple calls to be made from several API candidate. It uses Abstract Syntax Tree (AST) Evaluation to effectively measure function-calling abilities and identify specific model errors, such as incorrect function names, missing required parameters, or inappropriate data types.

### 5.3 RESULTS

We compare multiple models, including the GPT series and open-source models like Command R-plus and Open-Mistral 8x22B. These models have significantly more parameters than our 7B model. As shown in Table 4, InfCycle achieves performance comparable to these robust models. Notably, our approach significantly enhances the performance of small-size models, enabling Mistral (who struggles with the instruction following) to handle tool-calling tasks effectively. For stronger models LLaMA, our method improves its performance in complex scenarios involving multiple tool calls.

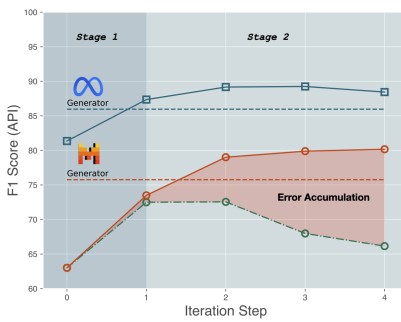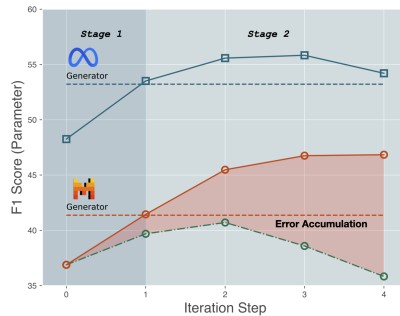

Figure 7: The figure illustrates the average API and Parameter F1 Scores on the human-annotated StableToolBench testset across different data synthesis iterations.

# 6  ANALYSIS

## 6.1  THE IMPACT OF DATA SCALING AND SYNTHESIS NOISE

**Is the *Simulator* necessary as the data synthesizer?**   As shown in Figure 4, models that integrate the *Simulator* with the *Generator* for iterative data synthesis demonstrate consistent improvements in performance. In the initial stages of synthesis, models often filter out substantial amounts of usable data due to their inherent difficulty in identifying correct execution trajectories. However, as the synthesized data grows, both Mistral and LLaMA effectively leverage this data, eventually outperforming models that rely solely on *Generator*-based synthesis.

**Is the Cycle Consistency mechanism necessary?**   We conduct iterative experiments without applying cycle consistency filtering. It can be observed that as the number of synthesis iterations increases, the performance gap gradually expands. This highlights the critical role of reliable verifiers in enhancing model capabilities through inference-time computation.

## 6.2  THE IMPACT OF DIFFERENT SYNTHESIZED DATA

We compare the differences in synthesized data between using Mistral and Qwen as the backbone for training the Mistral model. Figure 8 shows that the performance differences are not significant, indicating that variations in the quality of synthesized data from different models are insufficient to bridge the performance gap between them. This suggests that although the quality of synthesized data can influence training outcomes, the inherent capabilities of the model primarily determine performance in specific tasks.

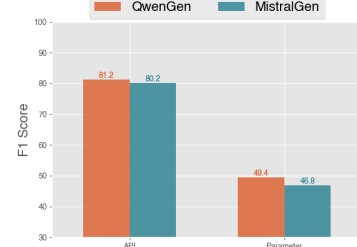

Figure 8: The average API and Parameter F1 scores for models trained on different datasets.

# 7  CONCLUSION AND FUTURE WORK

In this work, we enhance the model's tool use capabilities without relying on external supervision. Inspired by inference-time scaling, which increases the sampling space to enhance performance, this approach is particularly suitable for small models to facilitate self-improvement. We demonstrate that InfCycle effectively synthesizes high-quality data using the LLM as a proposer for inference sampling, and cycle consistency acting as process verifiers.

In the future, we plan to incorporate more tools and parameters into our research. This includes integrating various types of APIs and adjusting parameter scales to continuously enhance model performance. The trade-off between the number of parameters and inference cost remains a key focus, as it directly impacts the efficiency and applicability of our approach. Additionally, we plan to conduct more experiments to investigate further scalability factors.

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

# A DATASET AND EXPERIMENT DETAILS

## A.1 STABLETOOLBENCH DATASET

In this study, we use the StableToolBench Guo et al. (2024) environment and test set for method validation, rather than ToolBench Qin et al. (2023b). StableToolBench applies manual filtering and caching of HTTP requests to reduce API failures. As shown in Table 5, ToolBench has a significantly higher proportion of failed APIs, making StableToolBench more reliable for ensuring comparability and enabling a fair evaluation of model performance.

However, it's important to note that StableToolBench generates virtual request results using GPT-4, which can introduce biases during data synthesis. Therefore, we do not use the StableToolBench environment for synthesizing training data.

Table 5: This table illustrates the proportion of solvable data filtered out by StableToolBench.

| Benchmark | I1-Ins | I1-Cat | I1-Tool | I2-Ins | I2-Cat | I3-Ins | Sum. |
|---|---|---|---|---|---|---|---|
| ToolBench | 200 | 200 | 200 | 200 | 200 | 200 | 1100 |
| StableToolBench | 163 | 153 | 158 | 106 | 124 | 61 | 765 |

## A.2 HUMAN ANNOTATED STABLETOOLBENCH DATASET

For each user query, we conducted real-time access to the RapidAPI website and constructed the following testset, which includes both Single and Multiple types of user queries. As shown in Table 6, each sample includes the required API, corresponding parameters, and the access sequence.

Table 6: Statistics on the sample counts for different datasets (S represents Single samples, and M represents Multiple samples).

| Benchmark | I1-Ins (S) | I1-Cat (S) | I1-Tool (S) | I1-Ins (M) | Sum. |
|---|---|---|---|---|---|
| StableToolBench | 151 | 64 | 113 | 28 | 506 |
| | I1-Cat (M) | I1-Tool (M) | I2-Ins (M) | I2-Cat (M) | |
| | 36 | 46 | 22 | 46 | |

## A.3 BERKELEY FUNCTION CALLING DATASET

We also gathered data statistics on the Berkeley Function Calling in Table 7. This evaluation dataset primarily focuses on Python but includes other programming languages (such as Java and JavaScript), which increases the performance requirements for base models, as many models may not be proficient in languages beyond Python.

Table 7: This table shows the distribution of different types of data.

| Python | Java | JavaScript | Multiple | Parallel | Parallel_Multiple | Sum. |
|---|---|---|---|---|---|---|
| 400 | 100 | 50 | 200 | 200 | 200 | 1150 |

## A.4 TRAINING DETAILS

In our experiments, we utilize Mistral-7B-Instruct-v0.2, LLaMA3-8B-Instruct, and Qwen2.5-7B-Instruct as the foundation models, and the training process is conducted based on the alignment-handbook framework in a multi-round conversation mode. During the 1-epoch supervised fine-tuning (SFT) phase, we use a total batch size of 8, a learning rate of 7.0e-06, and a maximum sequence length of 4096. For the 1-epoch direct preference optimization (DPO) phase, we maintain a total batch size of 2, a learning rate of 5.0e-7, and a maximum sequence length of 1024, with the $\beta$ parameter set to 0.01. All experiments are conducted on a single machine equipped with 8 NVIDIA A100 GPUs, each with 40GB of memory.

Figure 9: The figure illustrates the steps in the *Generator* data synthesis process, showing how to transform API results into usable tool training trajectory data.

## A.5 GENERATOR AND SIMULATOR DETAILS

As shown in Figure 9, *Generator* sequentially executes three components: Query synthesis, Response synthesis, and Plan synthesis, in the data synthesis process. In each part, the model independently assesses plausibility to enhance trajectory accuracy. Given that we use a small-sized model, semantic checks may not filter out all errors. Effective filtering still relies on the cycle consistency mechanism.

As shown in Figure 10, we illustrate the potential intermediate results at each step of the Simulator. Unlike previous works like ReACT, which use historical information as the context for the current action, our approach relies solely on effective results from previous steps. This reduces the context length, ensuring the model is not constrained by long text processing capabilities. For multi-step tasks in StableToolBench, we summarize each plan into a Response, which serves as the final result.

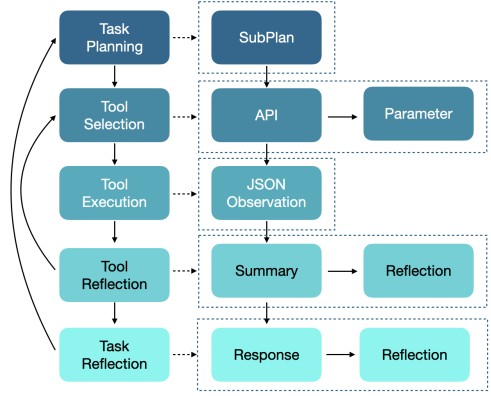

Figure 10: The figure depicts the Simulator's specific execution steps and intermediate results.

## A.6 PROMPTS

In this section, we present the key prompt templates we use in our data synthesis process, as shown in Figure 11, 12, and 13. We do not carefully select these prompts, as we focus on leveraging iterative synthesis techniques to generate data, rather than employing them for reasoning during inference. This approach emphasizes data generation while ensuring modeling flexibility.

**Prompt for the Generator to Generate Query**

**Simple Query**
You are a query generator tasked with creating realistic and natural queries based on a provided API call. Please generate a specific and complex user query based on the given API information.
Requirements:
- Realistic: The query should reflect what actual users might inquire about when trying to understand or utilize the function in real scenarios.
- Fluent: The query should be well-structured, clear, and free of grammatical errors.
- Parameter Reasoning: The generated query should demonstrate an understanding of the function's parameters, reasoning about how they should be correctly used or what their values should be.
Please don't mention API in the user query, but it needs to contain the parameters required to call the APIs.
Now generate query description for given API function call.
Input: function calling = {}

**Parallel Query**
You are a query generator tasked with creating realistic and natural query based on provided API call which necessitates multiple times using different parameter-value pairs.
The query should align with realistic user needs, structuring a scenario where the function's application is clearly required.
Requirements:
- Logical Flow: The query should be logically structured to necessitate multiple API calls, with the sequence of calls matching the order of parameters provided.
- Parameter Inclusion: Incorporate all necessary parameters into the query. The query should allow each paramerter for API call to be logically derived from the context.
- Clarity and Realism: The user query must be clear, grammatically correct, and realistically framed, resembling a genuine request that might prompt such an API interaction in a real-world application.
Please don't mention API in the user query, but it needs to contain the parameters required to call the APIs.
Now generate query description for given API function call chain.
Input: function calling = {}

**Multiple Query**
You are a query generator tasked with creating realistic and natural queries based on a provided API call. Using the provided list of APIs, select and combine given APIs to create a specific and complex user query.
Requirements:
- Establish Logical Relationships: Ensure that the API calls are logically related and form a coherent sequence. The query should reflect a natural flow where the output of one API informs the next, or where multiple APIs are combined to achieve a complex objective.
- Parameter Validity: Construct the query so that valid parameters for each API call can be inferred from the context. Ensure the query provides sufficient information to logically deduce the required parameters where applicable.
- Clarity and Realism: The user query must be clear, grammatically correct, and realistically framed, resembling a genuine request that might prompt such an API interaction in a real-world application.
Please don't mention API in the user query, but it needs to contain the parameters required to call the APIs.
Now generate query description for given API function call chain.
Input: function calling = {}

Figure 11: Instruction prompt for query generation.

---

**Prompt for the Generator to Generate Golden Trajectory**

**Simple Plan**
Given [User Query], [Function call] that can be used to solve the query, your task is to provide a concise, logical, and well-organized task plan centered around API function call to address the query
Provide the task plan without any "execution details", "result handling", or "error management".
***
Input:
[User Query]:{query}
[Function Call]:{function_call}
***
Please generate a plan in one sentence:

**Simple Answer**
Given [User Query], [Function call] with returned response used to resolve the query, your task is to generate a concise, coherent, and reasonable answer based on the available information from the function. Requirements:
- Ensure fluency and clarity: The answer should be well-structured and articulated in a clear, fluent, and natural manner.
- Match the function response: The answer must directly reflect the response of the function call. Avoid adding any outside knowledge or assumptions not provided by the function. - Correct details: The answer must fulfill the user's requirements and resolve the query satisfactorily. Now generate an answer for the given query and function call.
***
Input:
[User Query]:{query}
[Function Call]:{function_call}
***
Please generate an answer:

**Final answer** Given [User Query] and [Subtask with Subanswer], your task is to summarize the results of executing all subtasks and effectively consolidate these results to provide a comprehensive and accurate answer to the user query. The final answer should be detailed, complete, and well-structured, directly addressing the user query.
Requirements:
- Subanswer Utilization: The final answer must be based entirely on the subanswer, without incorporating any external knowledge or assumptions.
- Answer Resolution: Ensure that the final answer fulfills the user's requirements and resolves the query satisfactorily.
- Answer Quality: The final answer should be clear, detailed, and logically structured, providing a high-quality solution to the user query.
***
Input:
[User Query]:{query}
[Subtask with Subanswer]:{context}
***
Please generate a summary answer:

Figure 12: Instruction prompt for trajectory generation.

**Prompt for the Generator Semantic Checker**

**query check**
Please compare the given query with the target function call parameter and determine if they match. The evaluation criteria include:
- Query Clarity and Coherence: Assess whether the query is articulated in a clear, fluent, and natural manner.
- Parameter Derivability: Ensure that all parameter values required for the function call can be either directly extracted from the query or logically inferred based on the provided information. Please note that common sense knowledge can be used to infer parameters from problems.
If all the above criteria are met please first output YES/NO, and then give reasons for the judgment.
***
Input:
[query ]: {query}
[parameters]: {parameters}
***
Output:

**single answer check**
Please evaluate if the current answer effectively addresses the user query based on the information provided by the function response. The evaluation criteria include:
- Response Utilization: Check if the answer is obtained based on the given function response.
- Query Resolution: Determine whether the answer fulfills the user's requirements and resolves the query satisfactorily.
- Clarity and Coherence: Assess whether the answer is articulated in a clear, fluent, and natural manner.
If all the above criteria are met please first output YES/NO, and then give reasons for the judgment.
***
Input:
[query ]: {query}
[function response]:{function_resp}
[answer]:{answer}
***
Output:

**final answer check**
Please assess whether the given answer effectively solves the user's problem and whether the language is smooth, fluent, accurate, and concise. Please consider whether the answer responds directly to the query, is complete, and is clearly expressed.
If all the above criteria are met please first output YES/NO, and then give reasons for the judgment.
***
Input:
[query ]:{query}
[answer ]:{answer}
***
Output:

Figure 13: Instruction prompt for semantic checker

