# OpenReview forum: "InfCycle: Learning to Use Tools via Inference Compute and Cycle Consistency"
_ICLR.cc/2025/Conference — ICLR 2025 Conference Withdrawn Submission_

### Official Review · Reviewer_1Fb2 · 2024-11-04

**Soundness:** 3
**Presentation:** 3
**Contribution:** 2
**Rating:** 5
**Confidence:** 4

**Summary:**

This paper presents InfCycle, a framework aimed at improving Large Language Models' (LLMs) tool-usage capabilities for complex reasoning tasks. InfCycle leverages inference-time compute scaling, which boosts solution quality by increasing sampling during inference, and introduces cycle consistency as a process verification mechanism to ensure accurate data synthesis for model training. Key challenges addressed by InfCycle include the limitations of traditional methods like Proposer-Verifier, which often lack reliable verification for tool-based, multi-step reasoning tasks, and the difficulty in obtaining high-quality, large-scale training data in tool-use scenarios. InfCycle tackles these issues through three stages: (1) a Data Synthesizer Pipeline that collects real-world APIs, categorizes them, and generates user queries validated through LLM-based semantic checks; (2) Step-wise Cycle Consistency, which verifies the consistency of execution trajectories generated by the data synthesis steps to ensure logical coherence and semantic accuracy; and (3) a Multi-Stage Synthesis Strategy that uses A* search for efficient sampling, Direct Preference Optimization (DPO) to improve solution quality via pairwise comparisons, and iterative refinement to handle increasingly complex scenarios. Experiment results appear to show that cycle consistency as a verification mechanism overcomes the limitations of some existing methods.

**Strengths:**

- InfCycle allows LLMs to learn tool use independently, eliminating the need for external supervision or expert demonstrations.
- The cycle consistency mechanism provides a reliable verification process, overcoming traditional Proposer-Verifier limitations for tool use tasks.
- Through a multi-stage synthesis strategy with A* search and preference learning, InfCycle improves the model’s reasoning skills, particularly in executing multi-step tool actions and mitigating error accumulation.
- Evaluations on StableToolBench and the Berkeley Function-Calling benchmark show that InfCycle outperforms state-of-the-art baselines, achieving high pass and win rates—even with smaller model sizes.

**Weaknesses:**

- Although the authors highlight the efficiency of the data synthesis pipeline, InfCycle’s iterative multi-stage approach—combining synthesis, A* search, and preference learning—could still lead to substantial computational demands. The paper doesn’t analyze the computational costs associated with data generation in depth, which is crucial for assessing feasibility, especially in resource-constrained settings.
- The data synthesis process may also fall short in simulating the complexity of real-world tool use, where unpredictable errors or changing user needs often arise.
- Pass Rate and Win Rate are the main metrics used for evaluating model performance. However, these metrics may not fully reflect the quality of tool use, such as efficiency measured by the number of APIs called.
- The paper’s novelty is somewhat limited, as its multi-stage synthesis strategy appears similar to Chain of Preference Optimization (CPO) [1] , which fine-tunes LLMs to align each step of the chain of thoughts reasoning paths with those of tree of thoughts using the inherent preference information in the tree-search process. Furthermore, the step-wise cycle consistency mechanism overlaps with Step-DPO [2], which also treats each reasoning step as a unit for preference optimization.

[1] Chain of Preference Optimization: Improving Chain-of-Thought Reasoning in LLMs. NeurIPS'24.

[2] Step-DPO: Step-wise Preference Optimization for Long-chain Reasoning of LLMs.

**Questions:**

- Could the authors clarify how InfCycle’s approach specifically differs from Chain of Preference Optimization (CPO) and Step-DPO?
- Could the authors provide insights into the efficiency of the synthesized solutions? For instance, metrics on API call frequency or other indicators of solution efficiency would help to assess practical performance, complementing the Pass Rate and Win Rate metrics.
- Could the authors provide an overview of the computational cost involved in InfCycle’s multi-stage synthesis strategy?

---

### Official Review · Reviewer_qee4 · 2024-11-04

**Soundness:** 3
**Presentation:** 1
**Contribution:** 1
**Rating:** 5
**Confidence:** 2

**Summary:**

This paper considers the problem of data synthesis for tool usage in LLMs. The key motivation is that by leveraging tools, e.g., API calls, and other inference-time computation, LLMs are able to perform test time adaptation. The key idea of this paper is to have the LLM generate it's own API queries and then evaluate if those API calls were successful and semantically correct. This is used in two ways. First, in order to guide test time inference, e.g., via a tree of thought like search, as well as to act as preference data for DPO.

The results indicate that this approach is able to improve several models across a variety of tasks.

**Strengths:**

1. The problem of self-learning and in particular usage of tools for problem solving is relevant and of interest.
2. The experimental evaluation is thorough and does seem to mostly back the claims presented in the paper.
3. The key idea of generating synthetic data via self generated and evaluated api calls is clear and applicable to a large number of domains.

**Weaknesses:**

1. The biggest weakness of this paper is presentation. There are many ideas and they don't seem to flow well between each other. As a concrete set of examples:
   - Perhaps I'm missing something, but section 3 seems to be completely unrelated to the rest of the paper. The insight that process verifiers are better than correctness verifiers never seems to be build on.
  - I had no idea that part of the proposal involved fine tuning the LLM using DPO until page 6. In retrospect, reading the abstract and introduction, I could piece this together, but it does not come across at all during the first reading.

2. While the idea is effective, to me step-wise cycle just seems like another instantiation of the proposer/verifier split? Going back to presentation, cycle consistency is never formally defined, and so perhaps I'm misunderstanding, but this just seems to be checking the individual steps are correct?

**Questions:**

What *is* (in unambiguous formal language) cycle consistency? Please provide a definition!

I am open to raising my score, but at the moment, the understanding I've implicitly gleaned from the paper looks incremental.

---

### Official Review · Reviewer_Ew6c · 2024-11-04

**Soundness:** 3
**Presentation:** 3
**Contribution:** 3
**Rating:** 6
**Confidence:** 3

**Summary:**

This paper introduces InfCycle, a multi-stage data synthesis strategy for LLMs to improve reasoning and tool use through self-improvement without external supervision. It builds on the paradigm of proposer-verifier for inference scaling.  InfCycle utilizes cycle consistency for verifying intermediate steps, reducing error accumulation and enhancing data sampling efficiency. It incorporates a Generator and Simulator for generating training data and applies the A* search algorithm and preference learning to boost performance. Experiments on benchmarks demonstrate significant improvements, with Qwen2.5-7B outperforming GPT-4, achieving a ~75 pass rate and a ~79 win rate.

**Strengths:**

+ leveraging additional computation at test time to enhance tool use accuracy, enabling LLMs to improve without additional supervision

+ proposing a multi-stage approach using a Proposer and cycle consistency as a Verifier to boost data sampling and performance

+ showing significant performance gains, including Qwen2.5-7B surpassing GPT-4 on StableToolbench, with a 75.4% pass rate and a 79.6% win rate, and on  Berkeley Function Calling Benchmark where MetaLLaMA3-8B achieve an improvement of over 16.31 points.

+ analysis results show that process verifiers are more effective than solely outcome verifiers in enhancing long-range reasoning capabilities and in identifying high-quality execution trajectories.

+ multistep tool invocation is considered comprising planning, tool selection, execution, tool reflection and task reflection.

**Weaknesses:**

The experimental results are rather mixed. Why are the two benchmarks treated asymmetrically with respect to baseline and comparison models. Why not include results from GPT4-turbo. GTP4o-mini for both cases? Similarly, InfCycle uses 3 different models in first case (Table 2 and 3) but only 2 different models in second (Table 4).

**Questions:**

Could the authors explain the concerns about experimental evaluation in the weakness section?

---

### Official Review · Reviewer_W1Qa · 2024-11-04

**Soundness:** 2
**Presentation:** 2
**Contribution:** 2
**Rating:** 3
**Confidence:** 3

**Summary:**

This paper addresses the challenge of enhancing the reasoning abilities of large language models (LLMs) in integrating inference-time external tool invocations. Using a Proposer-Verifier approach, where LLMs receive automatic or human feedback on either the outcome or the sequential generation process, the paper introduces an iterative framework called InfCycle. InfCycle draws inspiration from the concept of cycle consistency in transformation tasks, where consistency implies that composing a transformation with its inverse keeps the output close to the identity map. Experimental results demonstrate that the proposed approach outperforms state-of-the-art baselines without relying on external supervision.

**Strengths:**

The paper tackles a timely and relevant  problem of improving LLM reasoning capabilities. The InfCycle framework is conceptually sound, and its key finding—that process verifiers are more effective than outcome verifiers—aligns with similar results in the literature. The experimental benchmarks (StableToolBranch and BFCL) and baselines are well-chosen and relevant to the problem. The results of the proposed framework, presented in Tables 2 and 3, are compelling across this comprehensive experimental setup.

**Weaknesses:**

The theoretical and conceptual novelty of the proposed framework is not sufficiently demonstrated in the paper. The main idea behind InfCycle and its adaptation of cycle consistency is not clear. In particular, the analogy to cycle consistency in this context is ambiguous, as the feedback from the simulator to the generator differs in type from the generator's input. Moreover, the lack of an overview example of the approach makes it difficult to understand specific contributions, such as the use of the A* algorithm and direct preference optimization.

**Questions:**

1. Could you clarify the concept behind InfCycle and its relation to cycle consistency with an example?
2. Could you also provide details on the application of Preference Learning and the A* algorithm in the paper using a simple example?

---

### Note · Authors · 2024-11-27

I have read and agree with the venue's withdrawal policy on behalf of myself and my co-authors.